# A Systematic Scoping Review of Pre-School Self-Regulation Interventions from a Self-Determination Theory Perspective

**DOI:** 10.3390/ijerph19042454

**Published:** 2022-02-21

**Authors:** Natalie Day, Fred Paas, Lisa Kervin, Steven J. Howard

**Affiliations:** 1Early Start, University of Wollongong, Wollongong 2522, Australia; dayn@uow.edu.au (N.D.); lkervin@uow.edu.au (L.K.); stevenh@uow.edu.au (S.J.H.); 2Department of Psychology, Education and Child Studies, Erasmus University Rotterdam, 3062 PA Rotterdam, The Netherlands; 3School of Education, University of Wollongong, Wollongong 2522, Australia

**Keywords:** self-regulation, intervention, scoping review, child-development, education, play

## Abstract

Self-regulation (SR) is considered foundational in early life, with robust evidence demonstrating a link between early self-regulation and longer-term outcomes. This has been the impetus for a growing body of intervention research into how best to support early SR development, yet approaches and effects are diverse, which complicates an understanding of the critical characteristics for effective early SR intervention. Using Self-Determination Theory (SDT) as a guiding framework, we present a scoping review of early SR-intervention research to identify the characteristics of pre-school interventions that show significant and strong effects on young children’s SR. Studies from peer-reviewed journal articles were included if they evaluated a SR intervention with pre-school children, were published between 2010 and 2020, written in English, and included a SR outcome measure. This yielded 19 studies, each reporting the efficacy of a different SR intervention. Results showed that content factors (what interventions do) interacted with their implementation (how, when, and by whom interventions are implemented) to discriminate the more versus less efficacious interventions. Through the lens of SDT, results further suggested that targeting *competence* through encouragement and feedback, and nurturing children’s *autonomy* distinguished more from less effective interventions. *Relatedness* was least able to discriminate intervention efficacy.

## 1. Introduction

### 1.1. Background

Self-regulation (SR) can be broadly conceptualized as the ability to control our attention, thoughts, emotions, and behaviors, despite competing distractions and impulses [1]. SR intervention research has gained traction in recognition of the short- and long-term outcomes associated with early self-regulation development [2]. Often considered a foundational ability in early life [3], early self-regulatory development predicts outcomes across the lifespan: SR in pre-school is positively associated with social competency, school engagement, and academic performance, and negatively related to poor long-term outcomes such as unemployment, depression and anxiety, criminal behavior, and alcohol and substance abuse [2]. While causality is not yet established from SR to these outcomes, these findings have been the impetus for a growing body of research into how best to support early SR development. It is largely accepted that children experience a period of rapid SR development between the ages of three and seven years, and the nonlinear way in which this occurs results in different growth trajectories [4]. Contributing to these disparities, both biological predisposition and environmental experience—which vary by child, family, and circumstance—influence early SR development [3]. Approaches to SR intervention, however, have been diverse and there is little consensus on the characteristics that are more or less conducive to SR growth.

While there has been a recent swell of early SR intervention studies, their approaches, implementation design, and effects have been varied. This has complicated an understanding of the critical characteristics of effective SR interventions. Without this, intervention design and implementation continue with little empirical guidance on which factors are more or less conducive to generating improvements in early SR. To address this issue, this scoping review sought to identify characteristics of (more, less, or in-) effective pre-school SR interventions from a Self-Determination Theory perspective (SDT).

In particular, this review focused on early interventions, in the preschool years, across various contexts (preschool/school and the home learning environment), and inclusive of intervention evaluation study designs, with the aim of reviewing the full breadth of contemporary approaches for SR intervention. While reviews will often focus on rigorous RCT designs in evaluating intervention efficacy, the current scoping review privileged the understanding of the breadth of approaches, evidence for their effects and commonalities across these, to provide insight into the diversity of approaches and implementation designs (rather than only those that have already progressed to large-scale RCT evaluation). 

A scoping review was identified as the most appropriate method to identify gaps in knowledge, and to scope the SR intervention literature [5,6], aligned with suggestions that this method of evidence synthesis is especially useful for gaining insight into intervention programs [7]. The broad questions that this review seeks to answer is what interventions exist, what their effects are, and whether there are commonalities among them. A scoping review, therefore, provides the most suitable solution for drawing narrative conclusions about the overall state of SR intervention research [8]. It was expected that findings from this review could be used to inform future research into creating, refining, and optimizing approaches to pre-school SR intervention, and also to inform the practices of those close to children. 

### 1.2. Conceptualisation of Self-Regulation

The ongoing and dynamic processes required for the self-regulation of cognition and behavior are influenced by internal states (emotion and cognition) and external influences (expected behavior and environmental stimuli [9,10]. While SR definitions, terminologies and operationalization are diverse [11], for the purposes of this review—and in line with conceptions that show strong prediction of later-life outcomes [12] SR is conceptualized as the child’s ability to modulate behavior toward achieving a goal, despite competing distractions. Modifying behavior towards a goal—the nature and importance of which are described in the sections that follow—is a complex process, however, consisting of several composite higher-order and lower-level skills. 

A prominent model of SR was proposed in context of Control Theory. Specifically, the feedback loop model proposed by Carver and Scheier [13] suggests there are three main elements to SR: *standards* (the ideals or goals that are self-selected); *monitoring* (comparing the behavior or current emotional state to the desired ideals, or goals); and *operate* (if there is a discrepancy between the current state and the goal, actions are determined and enacted to move the self forward from the current state to the desired state) [14,15]. Baumeister and Heatherton [16] elaborated the necessary *capacity* to move the self forward in the face of competing impulses or information; this capacity has often been taken to refer to the cognitive functions that underlie self-regulation, or executive functions (EFs) [17]. EFs, in this context, allow flexibility in thought and behavior, to adapt to new and changing circumstances and situations [18]. In this respect, EFs are necessary but not sufficient for successful SR, and EFs can also be deployed to activities other than SR [10].

‘Self-control’ is used synonymously with SR and is itself a contested concept. It is largely accepted, however, that self-control can be achieved in two ways: through the inhibition of certain behaviors or the initiation of other behaviors [19]. Responses that overcome immediate impulses differ to those that involve the initiation of goal-directed behavior; defined as inhibitory self-control and initiatory self-control, respectively [19]. Inhibition control is considered the suppression of goal-irrelevant stimuli and behavioral responses [20] with links to compliance [21] and over-regulation. Defined as a core EF [22], inhibition control does not align within the current conceptualization of SR and is therefore excluded throughout the selection process. 

In comparison, initiatory self-control (inhibition of impulsive behavior in favor of a long term, goal-oriented reward) finds alignment with the delay of gratification concept [23]. Considered a true test of self-regulation (with delay of gratification predicting similar patterns as SR for adaptive behavior in adulthood [24] evidence suggests that the delay is malleable [25,26,27,28]; and that there is a volitional (emotional) element to the regulatory behavior required for success in the test. This is not attributed to executive control alone but instead related to the goal-value and subsequent motivations toward successfully completing the task, which is supported by Self-Determination Theory (SDT [29]). 

Initiatory self-control and effortful control have similarities: encompassing our capacity toward regulated behavior [16], effortful control is described as the efficiency of executive attention and the *capacity* to inhibit an automatic response, in favor of an alternative response [30]. Considered a key aspect of SR, effortful control utilizes EFs to employ affective regulation to execute goal-directed behavior [31].

The distinction between inhibitory and initiatory self-control behaviors lies in the importance of ‘the goal’: self-regulatory efforts are those that actively engage in goal-directed behaviors, despite competing demands. This ‘management of the self’ for goal completion can be applied to four responses in which SR can be challenged: to regulate emotions; to regulate behavior; cognitive regulation; and social regulation. Any failure in one of the four responses is a threat to goal attainment and in this respect, the emotional, behavioral, cognitive, and social elements of SR are intertwined. SR, therefore, depends on the coordination of many processes that change developmentally over time [3] and is thus interpreted here as a multi-faceted construct that utilizes the previous four responses, collectively, toward goal-attainment. 

### 1.3. Self-Determination Theory as a Possible Framework for SR Change 

According to the feedback loop model, successful SR in real-world situations (which are complex and often emotionally laden) involves three elements: goal setting; motivation; and the capacity to persist toward goal attainment, all the while resisting competing impulses [32]. Yet this model provides little insight into mechanisms of the manner for inducing SR change. Self-Determination Theory (SDT), by contrast, represents one possible framework for explaining interaction and change in the content of the goal, motivation in the pursuit of that goal, and the regulatory processes through which the goal is pursued [29]. 

In consonance with SDT (or *Cognitive Evaluation Theory*, a sub-theory of SDT), a critical requirement for motivation toward goal attainment concerns the degree to which we are able to satisfy three basic psychological needs: autonomy, relatedness, and competence. Satisfaction of these needs, according to SDT, will promote intrinsic motivation [33], and thereby our curiosity and enjoyment of the task. This, in turn, creates optimal conditions for SR. 

As suggested by Cook & Artino [33], autonomy is nurtured through provision of choice and sensitive acknowledgement of feelings, and can be undermined through external rewards, negative judgement, and externally imposed goals. Competence is cultivated through optimal challenge and positive feedback, which contribute to a sense of ‘I can do it’ and avoids negative self-appraisals. Relatedness is supported through individuals and environments that provide a sense of genuine care and security, without criticism. SDT therefore focuses on the degree to which behaviors are enacted with a sense of volition [29], as opposed to heteronomy (a feeling controlled by either external actions or internal compulsions) [34] which may result in complaint behavior over regulated behavior. 

It seems logical that fulfilment of these needs can contribute toward supporting SR change among children. As a motivational theory, SDT is largely concerned with optimizing SR performance through identifying ideal conditions for SR, but with research suggesting that the provision of challenge is a pre-requisite for cognitive development [35]. Feelings of competence are enhanced through optimal challenge supported by positive encouragement [33] and thus SDT theory could also provide a framework for increasing one’s capacity for SR.

There is importance in impacting both in situ and long-term SR. Ideal conditions for SR (i.e., in situ, temporary change in SR) permit children to sustain attention toward SR-challenging experiences that can promote SR. Looking at Blair and Ursache’s [36] bi-directional model, for instance, cognitive control capacities (i.e., executive function) support children’s SR in a top-down manner, by controlling attention and thought under conditions of variable levels of arousal/reactivity. These stable capacities are often the targets of SR interventions seeking to change SR ability and related outcomes. Yet context-dependent factors such as emotional state and stress also exert ‘bottom up’ influences on SR—posing challenges to, or even hijacking, our ability for SR. Indeed, contextual factors have been shown to influence a child’s reactivity differently between contexts, and even activities in the same context, highlighting the unstable nature of SR performance [37]. Yet both acute SR and longer-term SR capacity are important. At manageable levels of arousal/reactivity, children are better able to engage with experiences that challenge their SR. This, in turn, is posited as a condition for stimulating growth in long-term and more stable SR abilities to cope better with a wider range of reactivity. While it is unclear from the current data whether SDT components affect the in situ and/or long-term capacity for SR, both are important conditions for learning and development of SR and beyond.

### 1.4. Previous Reviews

Previous reviews and overviews aim to synthesize literature in efforts to provide evidence for the effectiveness of SR interventions. Pandey et al. [38] conducted a systematic literature review and meta-analysis that included only randomized clinical trials across infancy to adolescence (from 0 to 19 years). The review sought to identify the effectiveness of universal SR interventions, as well as other SR-related outcomes (e.g., academic achievement and substance abuse). The meta-analysis reported SR improvement by intervention-type, i.e., consistent improvement was found in 76% of curriculum-based programs, in 67% of exercise-based programs, and in 50% of mindfulness and yoga programs. The study found that 34% of studies did not find noticeable change in SR, yet the finer-grained details of common implementation characteristics among these interventions by age (both for effective and non-effective characteristics) were not sought in this review.

In the formal education context, Dignath et al. [39] conducted a meta-analysis on self-regulated learning in elementary (primary) school, while Ursache et al. [40] drew associations between SR programs (based on a broad range of theoretical assumptions) and improvement in school readiness, with a specific focus on achievement. Both studies omit the importance of the home learning environment, and other associated non-formal contexts which have been shown to have substantial impact on children’s development [41]. This paucity of parent-based intervention research is recognized as such by Morawska et al. [42], whose overview identified only two parenting interventions for pre-school aged children. While positive and proactive guidance and parent involvement are shown to be essential in supporting children’s SR, the lack of evidence impedes generalizations.

Finally, Piquero et al.’s review [43] consolidated evidence of self-control interventions that were implemented with children under 10 years who demonstrated problems. The specific context and sample of children here prevents a direct association to typically developing populations.

Across these reviews, we do not have a clear sense of intervention characteristics that are critical for SR improvements during the pre-school period (aged three–five years). Given the heterogeneity of program models, it seems the next step in intervention research is to examine key criteria for effective (and non-effective) SR change across intervention types and context, inclusive of interventionist.

## 2. Materials and Methods

This scoping review was conducted in accordance with the Preferred Reporting of Items for Systematic Reviews and Meta-Analyses—extension for scoping reviews (PRISMA-ScR, see Appendix A) [44]. The protocol was developed and reviewed by members of the research team and registered with the Open Science Framework (https://tinyurl.com/2ukz6h2b (accessed on 22 December 2021)).

### 2.1. Eligibility Criteria

To be included in this review, studies had to report the results of a quantitative evaluation of an intervention that targeted pre-school children’s (mean age 3–6 years) self-regulation, and provide detail to link the intervention to SDT components, requiring sufficient detail about children’s activities. Studies were eligible for inclusion if they were peer-reviewed journal articles published between 2010 and 2020 (a previous review of studies published between 2000 and 2009 exists, of which only four of forty-nine studies reviewed were pre-school interventions; Pandey et al., 2018), written in English, included a SR outcome measure. Studies were excluded if the intervention did not have relevance to children’s everyday activities (for example, SR in the context of eating disorders), insufficient description of the intervention to determine alignment with the inclusion criteria, or if they characterized traditional EF measures as a SR outcome measure.

### 2.2. Information Sources

To identify relevant studies, a systematic search was conducted in early 2021 of the following five electronic databases: Scopus; PsycInfo; ERIC; ERC; A+ Education. This was supplemented by handsearching reference lists of articles that were included in the review, as well as searching names of SR programs that were explicitly mentioned within the articles.

### 2.3. Search Strategy

Search terms were developed, reviewed, and refined by the full research team. The following search terms were used to elicit a broad coverage of the extant literature: (“self-regulat*” OR “emotional regulat*” OR “behavioral regulat*” OR “self-control*” OR “emotional control”) AND (“intervention*” OR “training*” OR “program*”) AND (“preschool*” OR “kindergarten*” OR “early years” OR “reception” OR “childcare” OR “daycare” OR “early childhood education” OR “young children” OR “early childhood” or “nurser*”. Limitations placed on the search included a date range of 2010–2021, articles published in peer-reviewed journals, and in English only.

### 2.4. Selection of Sources of Evidence

Using the above search terms and strategy, 1197 results were returned. After removal of duplicates, this yielded an initial set of 922 articles, of which the lead author screened the titles, keywords and abstracts. Full texts were obtained and reviewed for studies that clearly met the inclusion criteria or where the relevance was unclear. A further five articles were identified from hand searching of references lists and programs mentioned. Two-hundred and thirty articles were taken to full-text screening. The lead author reviewed each of the full texts in relation to the inclusion criteria, while a second reviewer independently assessed a randomly selected 25% (*n* = 53) of these results. This independent review resulted in 96% agreement between the screeners. Disagreements were resolved by consultation with a third, and as a result all three reviewers agreed on the final determinations for selection of the articles after moderation. 

### 2.5. Data Charting & Data Items

The data charting and extraction were performed by the lead researcher, in an iterative and consultative process with the research team. Extracted were: characteristics of the article (author, year of publication, country); context (e.g., preschool, home); approach (e.g., child activities, physical/movement); intervention target/foci (SDT components); implementation design (interventionist, training, duration, and dose); outcome measures; and efficacy results (significance, effect size). Analysis of context as a potentially differentiating characteristic was not possible given that there was one study in the home learning context, one in the lab, and all the remaining studies in the formal learning context. For all studies that reported effect sizes or gave at least sufficient data from which to compute a Cohen’s *d* effect size, this was tabled both as an effect size statistic and a magnitude of effect (such that large = 0.8+, moderate = 0.5+, small = 0.2+ or no effect < 0.2; Cohen [45].

### 2.6. Critical Appraisal of Individual Sources of Evidence

Each of the 19 studies [1,46,47,48,49,50,51,52,53,54,55,56,57,58,59,60,61,62,63] included for final analysis were reviewed to ensure the data analyses performed were specifically and appropriately reporting intervention effects (i.e., growth in intervention group compared to a control condition). Two studies [64,65] were excluded from the analysis because intervention effects were not reported and could not be computed from the published data.

### 2.7. Synthesis of Results

Consideration of Self-Determination Theory (SDT) was not a requirement for studies to be included in this review; rather, SDT was used as a framework through which to classify interventions and as an initial framework to account for differences in intervention effects. The studies were initially grouped by SDT categories specifically, the inclusion of intervention components explicitly and intentionally targeting autonomy, relatedness, and/or competence [29]—in line with the theoretical framework guiding this review.

Interventions were deemed to target autonomy if they featured one or more of the following features: shared control, child-led activity, and fostering choice [33,66,67].

Interventions were deemed to include ‘relatedness’ if the interventionist played a significant recurring role in the children’s lives, such as the teacher or parent. As defined by Niemiec and Ryan [68], relatedness is cultivated through genuine connection and a sense of belonging in the child, and this connection is more likely to be readily established with parents and familiar teachers because there is an existing basis of mutual connection in both the family and school community. While relatedness can be fostered by other adults (e.g., psychologists), adults unfamiliar to the child would need to invest energy to cultivate and foster feelings of relatedness. Unknown instructors or researchers were therefore not viewed as offering the same degree of relatedness, unless this was specifically and explicitly targeted by the intervention.

Interventions were deemed to target competence if they exposed children to self-regulatory challenge, and provided encouragement or feedback [33]. This challenge could be applied to any one of diverse contexts to be considered as targeting child competence: cognitive (attention or persistence), behavior (impulsive behaviors, delay of gratification), social (social conflict), and/or emotional (frustration).

Within these categories, studies were considered for statistical significance and effect size (large, medium, small, no effect). In this way this review sought to survey the evidence for (effective) approaches to employing SDT principles in support of early self-regulation, and thereby bring to light current knowledge and gaps regarding intervention characteristics that might be more or less efficacious.

## 3. Results

The search strategy and inclusion criteria yielded 19 studies from which data items could be extracted. The full screening procedure and results can be seen in Figure 1. The 19 studies included in this review comprised 4177 participants with a grand mean age = 54.5 months (SD = 8.5). Two studies did not publish children’s ages. A total of 19 *p*-values and 19 effect sizes could be extracted. Descriptive information from tabling procedures is presented in Appendix A. The samples were from United States (*n* = 11), Canada (*n* = 3), Australia (*n* = 1), Italy (*n* = 1), the United Kingdom (*n* = 1), Turkey (*n* = 1), and Taiwan (*n* = 1).

Applying the planned organization of studies by SDT components, interventions explicitly integrated the following elements (either in isolation or with one other SDT component) at the following frequencies: relatedness (six studies: four in isolation, one with competence, and one with autonomy); competence (nine studies: two in isolation, six with autonomy, one with relatedness); autonomy (seven studies: none in isolation; six with competence, and one with relatedness), and those that include all three (five studies). The number of studies and overall pattern of results (i.e., significance, effect size) is provided at Table 1. Results are organized by each SDT component, either alone or in concert with another, and studies were further explored by extrapolated characteristics as detailed in Appendix A.

### 3.1. Relatedness, Alone or with One Other SDT Component

Interventions that explicitly targeted relatedness—either on its own (*n* = 3) or with another component (*n* = 2)—were those in which the interventionist was either a class teacher or a parent. Interventions that were led by visiting instructors, temporary teachers, or researchers did not specifically target this component (per previous criteria outlined).

Interventions targeting relatedness were the least successful in terms of efficacy, with only three of five studies returning significant effects, and only one of these having an effect size that was not small. Of the three studies that were significant, two used mindfulness-based activities as their approach, and the other used child activities. For the two non-significant studies, the approaches were child activity and a nature-based program.

Interventions that simply and exclusively (in relation to SDT components) employed a familiar adult as the interventionist (thereby leveraging a sense of relatedness) had mixed evidence. Two of the three studies achieved significance, one with a moderate effect [54] and the other small [50]—both were mindfulness programs. In mapping program characteristics with evidence of efficacy, interventionist training appeared to vary in concert with effects. For instance, Taylor and Butts-Wilmsmeyer’s [58] nature-based program did not include interventionist training or professional development for the educators. The other inherent constraint to these studies is that they only implicitly sought to foster relatedness and did not do this explicitly or in conjunction with any other SDT component. There were also differences in dosage and duration, with the non-significant intervention implementing the program at a lower dosage (not daily as per the significant studies) and for less time (one semester versus a minimum of six months by the significant studies).

Combining relatedness with autonomy, Jelley et al. [51] yielded a small, significant effect size through offering shared control between parents and children when completing home-based activities which were sent via an app. The activities and games were designed so that SR is required to complete them.

The final study, which combined relatedness and competence, was non-significant.

### 3.2. Autonomy, Alone or with One Other SDT Component

No intervention explicitly targeted autonomy in isolation of other SDT components. Yet there were interventions in which autonomy was targeted in concert with one other SDT component (*n* = 7). In these cases, the autonomy component of the interventions included at least one, or a combination, of the following three elements: giving children choice; shared control between adults and children; and child-led activities. Six of the interventions (86%) returned significant effects.

Interventions were diverse in their dose (ranging from 40 min a week to daily practice) and duration (M = 8.9 weeks, SD = 4.9, ranging from 3 to 18 weeks). Intervention approaches were less diverse, with studies predominantly utilizing child activities to foster SR growth (*n* = 6), while one used a movement-based program. In all but one case (Jelley et al., 2016), the interventionist was unknown to the children—either a trained professional (mindfulness instructor), researcher, or unknown teacher.

Among the intervention evaluations that included autonomy and returned significant effects, two that paired intervention components to also promote competence (i.e., challenge, encouragement, and feedback) achieved large effects. The first of these is Robinson et al. [55] who implemented an intervention that took a mastery approach to physical movement, targeting children’s intrinsic motivation and persistence. The children self-selected the activities, self-monitored their progress, and received personal, meaningful evaluations and adult-feedback. The second is Sezgin and Demiriz [61] who implemented a program that encouraged children to make plans and decisions, lead, demonstrate effort to complete challenging tasks, and exposed children to new situations.

Of the remaining significant studies, small effects were achieved. In all cases, not all components of autonomy were included. For instance, children in the Red Light, Purple Light (RLPL) interventions by Duncan et al. [63] and Schmitt et al. [56] were only given opportunities to share control through leading circle-time games (not choice), while Shiu et al.’s [62] program gave children choice and opportunities to lead in story-telling activities (not shared control). Yet another commonality of these studies was that they also included the challenge aspect of competency, although did not explicitly mention use of encouragement and feedback. The RLPL study by Schmitt et al. [56] sought to extend and replicate the earlier the work of Tominey & McClelland [59] that did not achieve significance. Two implementation limitations of Tominey & McClelland’s work were identified and modified for Schmitt et al.’s [55] study: first, that children were not assigned to intervention or control classrooms, but rather randomly assigned to condition; and second, the intervention occurred outside of the classroom. Schmitt et al. [55] therefore evaluated the RLPL program as a classroom-based intervention. Similarly, Duncan et al. [63] replicated Schmitt et al.’s [55] design, with both yielding significant and small effects.

The one intervention that paired autonomy with relatedness yielded a small effect. This study [51] similarly featured only partial autonomy features. While these results suggest potential efficacy from targeting child autonomy in early SR interventions, the lack of autonomy-only interventions made it difficult to discern if comprehensively targeting autonomy was sufficient to generate a meaningful, positive effect. Alternatively, these effects may be more highly influenced by their strategies to foster competence (e.g., encouragement and feedback), or whether both autonomy and competence should be combined.

### 3.3. Competence, Alone or with One Other SDT Component

Interventions that targeted competence alone (*n* = 2), or with another component (*n* = 7), were those that created self-regulation challenges for children and/or provided encouragement and feedback. Of the nine intervention evaluations, 78% (*n* = 7) were significant with effects that ranged from small to large.

Interventions were again diverse in dosage (ranging from 40 min a week, to 20–30 min of daily practice) and duration (M = 10.2 weeks, SD = 6.1, ranging from 3 to 24 weeks). The most frequent approach used was child activities (*n* = 6), followed by mindfulness programs (*n* = 2), and a movement-based program. In all but one case [1], the interventionist was unknown to the children.

Interventions and evaluations that found moderate to large effect sizes were found when programs intentionally and explicitly provided encouragement and feedback to children, either with and without facets of autonomy. For instance, both Robinson et al. [55] and Sezgin and Demiriz [61] featured these components of competence, as well as all three of the components of autonomy, yielding large effects. Yet Poehlmann-Tynan et al. [53] and Flook et al. [49] targeted only encouragement and feedback, in the absence of challenge and autonomy, and also elicited large and moderate effects, respectively. Both Poehlmann-Tynan et al. [58] and Flook et al. [59] used a program called Kindness Curriculum, a mindfulness-based sequence of lessons. One example of encouragement and feedback provided was through the active engagement element of the lessons: Lesson 1 of the program concerned awareness and paying attention. In the active engagement section of the lesson the children practiced feeling their breath as they felt their bellies get bigger and smaller. There were explicit instructions for the instructor to check-in and support the children to see what they noticed about their breath and assigned labels to it. These results are suggestive of the particular importance of these strategies to build competence.

Interventions that generated a significant, yet small effect size included SR challenge, although without explicitly or necessarily targeting feedback or encouragement [56,62,63]. As above, this pattern was found when some features of autonomy were also included in the intervention, and also when autonomy was not an explicit target of the intervention.

Further supporting this pattern, two studies did not achieve significance. Howard et al.’s [1] intervention commenced with online professional development videos, and then provided adult practices and playful child activities that specifically targeted self-regulation. In relation to its competence targets, activities were designed to explicitly include the need for self-regulation and included instructions to increase complexity as children’s proficiency increased. It did not, however, explicitly target, prompt, or require specific forms or frequency of encouragement or feedback. Similarly, Tominey and McClelland’s [59] Red Light, Purple Light program used challenging child activities without explicating the need for or form of feedback or encouragement.

### 3.4. Competence, Autonomy, and Relatedness Combined

Featuring all three SDT components, five studies demonstrated mixed results. Only two of these were significant, yielding large [48] and small effects [47], both of which included all elements of all three SDT components. Both significant studies were curriculum-based interventions (Tools of the Mind, *Tools*), yet a third *Tools* study [57] did not return significant results. Comprised of the same SDT components, Solomon et al.’s intervention does not fit the previous patterns of the data that suggest the importance of encouragement and feedback for significant and sizeable effects. In evaluating differentiating factors between the studies, it is notable that both the significant studies used unblinded teacher reports of classroom behaviors, whereas Solomon et al. [57] used a more objective task-based Head-Toes-Knees-Shoulders. It is unclear whether teacher reports may have introduced unintended biases into the intervention effect’s estimates.

The remaining two non-significant studies were those by McClelland et al. [52] and Meuwissen and Carlson [60], whose programs also targeted challenge (not encouragement and feedback). While McClelland et al.’s intervention shares competence and autonomy SDT components with the other RLPL studies (discussed in the last section, given that relatedness was not leveraged or targeted), this instantiation was delivered by the teacher and returned a non-significant intervention effect. Additionally, this study administered activities for 30–40 min per week, compared to 40–60 min per week [55] or 30 min of daily practice [54] in the other RLPL evaluations. Meuwissen and Carlson’s [57] study included content features that might suggest positive effects, on the basis of the patterns across these studies, this program was implemented in just one lab session and was thereby highly constrained in dose.

## 4. Discussion

Using SDT theory as a framework for this scoping review of early SR interventions, results suggested that the SDT characteristic associated with higher efficacy was competence, especially when encouragement and feedback was provided. Challenge, the second aspect of competence, was also present in efficacious studies, although did not appear to account as strongly for the strength of the effect. Autonomy also featured in a number of effective interventions, but there was insufficient evidence to suggest its particular impact on SR development. Less evidence was found to support the SR effects of relatedness; in isolation, a known interventionist such as a parent or teacher is not enough to generate successful intervention results. This may differ in cases of interventions that specifically target (rather than leverage) this component.

Research suggests that challenge is necessary for cognitive enhancement [35], and cognitive gains are seen when difficulty increases. The results of this review extend this, however, and instead promote a more comprehensive view of competence as a more critical feature for SR growth. Programs such as Red Light, Purple Light designed child activities that increased in challenge, yet did not return sizable SR gains. Given that these games are played in a group setting the level of challenge may not be suited to the abilities for all children to elicit SR gains. Challenge may therefore need to be individualized and meaningful.

A reason why challenge in isolation may not enhance SR is that when children experience challenge, they face dysregulation and are better placed to overcome competing distractions through adequate adult support (co-regulation) to further their SR. In this way, the adult supports the emerging self-regulatory skills of the preschooler so that their ability to self-regulate grows over time [69]. This notion draws on seminal theoretical and empirical research which highlights the importance of a more skilled ‘other’ in learning [70]. This review demonstrates that in interventions that were inclusive of challenge but not encouragement and feedback, SR change was less than those that included both.

Socio-cultural theories lay heavy emphasis on interaction through speech and dialogue in shaping thought, accentuating that self-regulation development occurs through internalization of language-based interactions with others [71], and thus the quality of encouragement and feedback as a form of co-regulation is important.

Theories of motivation and praise [72] demonstrate that praise for effort leads children to adopt a mastery approach toward challenging tasks (as demonstrated by Robinson et al. [52] in this review), while exhibiting pleasure, interest, and enthusiasm [73] toward goals that generate learning; and is thus aligned with intrinsic regulation as per SDT. Conversely, person-centered praise (focusing on character traits over effort) is linked with challenge avoidance after children experience failure and leads children to choose goals that maintain an easier level of success, linked to extrinsic motivation. While SDT proposes the ideal conditions for optimal intrinsic (self) regulation, the information presented thus far suggests the theory can be extended, in conjunction with other theories to inform how we could further develop SR in context for preschoolers.

With competence at the core, features of autonomy and relatedness can be utilized to consolidate theories presented thus far. While autonomy is conceptualized through the lens of SDT as the provision of choice, shared control, and child-led activity, autonomy may also exist through the resolution of dysregulation. Autonomy-supportive behaviors (from adult caregivers) encompass both autonomy and competence components of SDT, and includes providing children with choices, helping them to identify mistakes themselves, and encouraging children to take the initiative [67,74,75], each of which can support children’s SR development [60,66,76,77,78] when adults optimize the opportunities to do so. Hammond et al. [79] propose how this might be achieved in practice with a contingent 4-tier hierarchical process beginning with sensitivity (e.g., encourage, sustain attention, goal reminder), through to physical direction. In this way, the child’s autonomy remains intact when they experience dysregulation, and caregiver support is contingent on a child’s responses rather than providing scaffolding that pre-empts and reduces challenge [80] and consequently limits SR development.

Feelings of autonomy and volition are perhaps more important than the origin of autonomy (whether it is intrinsic by task choice or external through teaching practice, see [70]) and thus understanding how children perceive and demonstrate agency is also worthy of investigation in this area of research. Suggestions made in this review offer an additional way autonomy may be nurtured, through autonomy support behaviors. Undermining children’s autonomy may negatively impact their ability to regulate effectively, and hinder SR development. Given SR is conceptualized as a goal-directed behavior enacted despite opposing distractions, the value of the goal is important. Using Cook & Artino’s [33] illustration, external influences in a child’s everyday experiences (e.g., adult-directed tasks, rule-enforced games) impact how the goal is valued, resulting in extrinsic motivation. By allowing children the autonomy and agency to engage in making meaningful choices in relation to tasks/activities (e.g., what, who with, how, and for how long), internal motivation will be facilitated and the child more likely to be intrinsically regulated.

Play is one avenue for how children can be afforded feelings of autonomy. A recent evaluation of the literature suggests that play can be conceptualized as unfolding across a continuum on which child autonomy varies [81], from free play characterized by full child-autonomy (under which pretend play might fall; Pyle & Daniels [82]) to games that limit autonomy through embedded game rules, and then to direct adult-led instruction in the absence of child autonomy. Play can be influenced by the context in which it occurs: the structures in place allow the extent to which children are able to use their agency [83], as well as available resources across a wide spectrum; from adults and other children to imagination and tools [84]. Play is a context, therefore, which offers an opportunity for the components of SDT to be fulfilled.

Through the lens of Attachment Theory, the importance of relatedness for children is seen. Defined as a deep, affectionate, connecting bond between people, parent–child attachment is associated with outcomes such as acceptability of challenges, social competence, and emotional regulation [85] (see Bergin & Bergin [86] for an overview of the literature), all of which require self-regulatory abilities. Securely-attached children, those who experience a deep sense of genuine care (as per SDT) are more likely to explore freely and are more willing to confront challenges [86] than children who are not secure. In the formal learning context, educators must also connect with children to parallel secure parent-attachment so that the adult is used as a safe base from which the child can explore and tackle challenge, knowing the adult is there for comfort and reassurance [87]. For SR development and growth, a safe and secure environment is therefore necessary for a child to encounter challenge, and to be supported through overcoming distracting impulses when dysregulated. In cultivating this environment, parents have the upper hand over educators, and while teachers can cultivate positive relationships, the extent may not fulfill the same requirements in which children perceive a sense of relatedness with a parent. Research on the consistency of early attachment is mixed, pointing to that while attachment relationships may remain constant throughout childhood, sometimes they change [88], pointing to the plausibility that adult-child relationships can be shaped and influenced by implementing behaviors that help to form secure attachments.

The review also highlights a paucity in how little intervention research is carried out in the home learning environment with parents, with only two of the nineteen studied being parent-based interventions. The home learning environment, and associated experiences, have been identified as among the most important influences on children’s development [41,89,90,91,92], yet there remains little in the way of practical guidance for parents to endorse SR-promoting behaviors, as supported by this scoping review. Using parents alone is not enough to elicit impactful results however, according to the findings of this review. Evidence suggests praise mechanisms may also operate in the home learning environment [93], a context where children can also experience a more individualized attention, and thus parent-based interventions are a plausible and possibly fruitful avenue for future intervention research given the head-start parents have in creating a sense of relatedness. Information discussed thus far points to the home learning environment as a profitable context for intervention research for the following reasons: it affords the attention and knowledge required to design individualized challenging experiences for the child; one-on-one autonomy support and associated encouragement and feedback can be provided by the caregiver; it allows the child to make meaningful choices; and it occurs alongside an adult with whom the child has a deep, affectionate bond, from where the child feels secure and safe to take risks.

Based on the findings of this review and theoretical connections, we advocate for settings in which children can experience feelings of competence, autonomy, and relatedness in their everyday activities through play. Results of this review imply that interventions would do well to consider nurturing environments that offer child-initiated and autonomous activity, wherein children are more likely to be intrinsically motivated. Interventions that include tasks that offer children experiences of frequent and appropriate challenge allow for caregivers to provide co-regulation through contingent autonomy support, during which adult encouragement through praising effort toward overcoming challenges can be implemented. Future work to explore how these components can be implemented is needed to further elaborate on this. In particular, future research is required to evaluate: how best to approach intervention in the home context; if parents can be successfully taught to enact autonomy supportive behaviors in relation to children’s SR; the ways autonomy and relatedness might support competence in an intervention context; the intervention components that generate greater effects; and which choices are ‘meaningful’ for children in the play context. This research highlights the need for children to experience and engage with meaningful and appropriate self-regulatory challenge, yet further research is required to understand how to tailor programs to ensure children experience optimal levels of challenge in relation to their current SR needs.

### Limitations

Our scoping review has some limitations: one of these concerns is the use of grey literature. Grey literature can offer scoping reviews the benefit of reducing publication bias and offering contextual information of ‘in the field’ practices [94]. This, for instance, may shed light on how SR is perceived by early childhood educators [95] and the nature of current SR practice. Interventions using different conceptions or terms for SR may yield useful insights, but were not captured by this review (e.g., executive function interventions). At the same time, they could have ‘muddied the waters’ for what is already an eclectic evidence base.

Another limitation of scoping reviews is the qualitative synthesis of findings across discrepant designs, methods, and interventions. This requires a theoretical frame by which to classify, synthesize and interpret findings, although other theoretical frameworks that better account for patterns in the data are possible. While this is a perpetual limitation of scoping reviews, it is notable that the current theoretical framework appeared to account well for the patterns in these studies’ results. The inability to meta-analytically synthesize the evidence is also a limitation of scoping reviews yet doing so would negate the benefits of insights obtained from the diverse and heterogenous nature of studies explored by scoping reviews.

Interventions that could not be classified according to SDT (e.g., [96,97]) could not be considered given the aim of this review. However, this leaves open the possibility that other approaches might be equally (or even more) effective, or that components that increase effectiveness might differ for other approaches (e.g., professional development approaches that do not specify child activity, but rather intervene within it). Still, this review provides important scoping of this field of research, on which future interventions and reviews can be based.

## 5. Conclusions

In relation to the research question guiding this review, the findings suggest that content factors (what interventions did) and implementation design factors (how interventions executed them) interacted to account for the differences in the presence and size of intervention effects. Results pointed to competence being a core driver for SR growth, yet it is unclear whether studies allowed children perform to their best SR abilities, or whether they measured growth in SR. While the review has thus far assumed that autonomy may play a supportive role alongside competence or relatedness, it may be the case that the effects are not additive. Various combinations may increase or decrease efficacy in a non-additive manner, for instance, with a single SDT factor generating additional benefit when combined with one or more other factors. The fact that no single study included autonomy in isolation means we cannot draw conclusions as to the effect of this component on its own. This possibility requires further research to investigate. The home learning environment, particularly, is an under exploited opportunity for pre-school SR development. This systematic scoping review presents a first step toward understanding the intervention components and implementation features that are more likely to yield effects in children’s SR.

## Figures and Tables

**Figure 1 ijerph-19-02454-f001:**
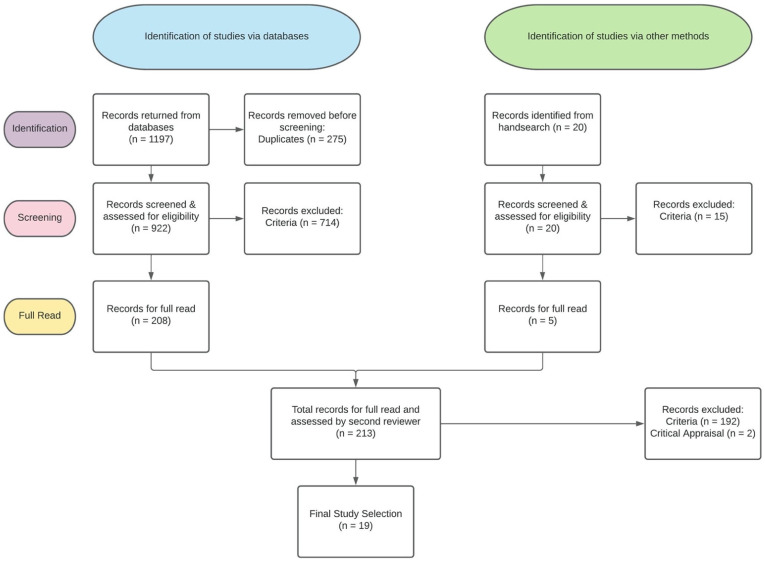
Diagram to illustrate study retrieval and selection process.

**Table 1 ijerph-19-02454-t001:** Table showing the frequency of the three SDT components integrated in interventions, and their corresponding effect sizes (ES).

Categories ^1^	Total no. of Studies (*n*)	Significant (*n*)	Large ES (*n*)	Moderate ES (*n*)	Small ES (*n*)	No Effect (*n*)
R	3	2	0	1	1	1
RC	1	0	0	0	1	0
RA	1	1	0	0	1	0
C	2	2	1	1	0	0
CA	6	5	2	0	3	1
A	0	0	0	0	0	0
CRA	5	2	1	0	3	1
None	1	0	0	0	0	1
Total R	5	3	0	1	3	1
Total C	9	7	3	1	4	1
Total A	7	6	2	0	4	1

^1^ R = relatedness, A = autonomy, C = competence.

## Data Availability

Appendix A is available in Appendix A.

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
