# Peer review of "A Systematic Scoping Review of Pre-School Self-Regulation Interventions from a Self-Determination Theory Perspective"

_ijerph, 2022, doi:10.3390/ijerph19042454_

Round 1
Reviewer 1 Report
This manuscript is a scoping systematic review of early self-regulation intervention using the self-determination theory as a guiding framework. The authors argue that scoping review has advantages over traditional systematic review because it included more diverse intervention contexts and methods rather than strict RCT. I commend the clear conceptualization of self-regulation, especially the effort of including both the top-down/cognitive and the bottom-up/emotional and physiological processes. The developmental significance of self-regulation is also pointed out in the introduction. I completely agree that it is important to include contexts beyond the school environment, and family and other out-of-school contexts play important roles in shaping self-regulation development. However, my concerns about theorization as well as the analysis make me unconvinced by the result.
The authors argued that according to SDT, SR change can be induced through satisfying the needs of autonomy, relatedness, and competence. It was specified that fulfilling the three psychological needs can promote intrinsic motivation. This argument makes sense when we talk about conditional change in SR. That is, a child can temporarily perform better in a context where the three needs are fulfilled, but changing the condition would also change the SR performance. Such a change in SR is temporary and reversible. However, the goal of intervention is long-term changes in SR regardless of condition and context. I recommend the author elaborate on how conditional change in SR can lead to long-term change in SR as a developmental outcome.
The author pointed out that previous reviews failed to include SR intervention in out-of-school contexts. However, the synthesis of the results only included analysis of the autonomy, relatedness, and competence components. To echo the study rationale, analysis of the intervention contexts should also be included. The role of the three psychological needs can potentially be moderated by the intervention contexts.
I am also worried that the search terms used for the search were not inclusive enough and many relevant studies were not included. For example, self-regulation during preschool years is often conceptualized and measured as executive functions and school readiness, but these two terms were not included. Subcomponents of executive functions (e.g., inhibitory control and cognitive flexibility) are sometimes used as the outcome in the studies without explicitly using the word “regulation”. For example, Head Start REDI and the Chicago School Readiness Project are two intervention programs that target preschoolers’ self-regulation, some of the publications even tested longitudinal outcomes in follow-up studies. Moreover, Head Start REDI included home components in later development. These two intervention programs use several curriculums, including Tools of Mind, The Incredible Years, and the PATH. Here are some publications between 2010 and 2020 that are relevant but not included:
Raver, C. C., Jones, S. M., Li-Grining, C., Zhai, F., Bub, K., & Pressler, E. (2011). CSRP's impact on low-income preschoolers' preacademic skills: Self-regulation as a mediating mechanism. Child Development, 82(1), 362-378.
Jones, S. M., Bub, K. L., & Raver, C. C. (2013). Unpacking the black box of the Chicago School Readiness Project intervention: The mediating roles of teacher–child relationship quality and self-regulation. Early Education & Development, 24(7), 1043-1064. doi:10.1080/10409289.2013.825188
Li-Grining, C., Raver, C., Jones-Lewis, D., Madison-Boyd, S., & Lennon, J. (2014). Targeting classrooms' emotional climate and preschoolers' socioemotional adjustment: Implementation of the Chicago School Readiness Project. Journal of Prevention and Intervention in the Community, 42(4). 264-281.
Zhai, F., Raver, C., Jones, S. M., Li-Grining, C. P., Pressler, E., & Gao, Q. (2010). Dosage effects on school readiness: Evidence from a randomized classroom-based intervention. Social Service Review, 84(4), 615-654.
Zhai, F., Raver, C. C., & Li-Grining, C. (2011). Classroom-based interventions and teachers’ perceived job stressors and confidence: Evidence from a randomized trial in Head Start settings. Early Childhood Research Quarterly, 26, 442-452. doi:10.1016/j.ecresq.2011.03.003
Zhai, F., Raver, C. C., & Jones, S. M. (2012). Academic performance of subsequent schools and impacts of early interventions: Evidence from a randomized controlled trial in Head Start settings. Children and Youth Services Review, 34, 946-954. doi:10.1016/j.childyouth.2012.01.026
Bierman, K. L., Nix, R. L., Heinrichs, B. S., Domitrovich, C. E., Gest, S. D., Welsh, J. A., & Gill, S. (2014). Effects of Head Start REDI on children's outcomes 1 year later in different kindergarten contexts. Child Development, 85(1), 140-159. doi:10.1111/cdev.12117
Nix, R. L., Bierman, K. L., Heinrichs, B. S., Gest, S. D., Welsh, J. A., & Domitrovich, C. E. (2016). The randomized controlled trial of Head Start REDI: Sustained effects on developmental trajectories of social-emotional functioning. Journal of Consulting and Clinical Psychology, 84(4). 310-322.
Sasser, T. R., Bierman, K. L., Heinrichs, B., & Nix, R. L. (2017). Preschool intervention can promote sustained growth in the executive-function skills of children exhibiting early deficits. Psychological Science, 28(12), 1719-1730.
Bierman, K. L., Welsh, J. A., Heinrichs, B. S., Nix, R. L., & Mathis, E. T. (2015). Helping Head Start Parents promote their children's kindergarten adjustment: The research-based developmentally informed parent program. Child Development, 86(6), 1877-1891. doi:10.1111/cdev.12448
Bierman, K. L., Heinrichs, B. S., Welsh, J. A., Nix, R. L., & Gest, S. D. (2017). Enriching preschool classrooms and home visits with evidence-based programming: Sustained benefits for low-income children. Journal of Child Psychology And Psychiatry, 58(2), 129-137.
Moore, J. E., Cooper, B. R., Domitrovich, C. E., Morgan, N. R., Cleveland, M. J., Shah, H., & ... Greenberg, M. T. (2015). The effects of exposure to an enhanced preschool program on the social-emotional functioning of at-risk children. Early Childhood Research Quarterly, 32, 127-138. doi:10.1016/j.ecresq.2015.03.004
Shah, H. K., Domitrovich, C. E., Morgan, N. R., Moore, J. E., Cooper, B. R., Jacobson, L., & Greenberg, M. T. (2017). One or two years of participation: Is dosage of an enhanced publicly funded preschool program associated with the academic and executive function skills of low-income children in early elementary school? Early Childhood Research Quarterly, 40, 123-137.
Due to the omission of relevant studies and the small number of included studies, I feel not convinced by the synthesis of the results. With only 19 studies in total and small numbers of studies in each category, it is a stretch to say that there is a “pattern” when more “competence-targeting” studies demonstrated significant results. Moreover, some studies included more than one component, and the effect of the inclusion of components may not be additive. The combination of certain components may yield a desirable result while each component cannot produce an effect on its own.
Author Response
Responses to comments of Reviewer 1 (Reviewer comments are printed in boldface)
..my concerns about theorization as well as the analysis make me unconvinced by the result.
The authors argued that according to SDT, SR change can be induced through satisfying the needs of autonomy, relatedness, and competence. It was specified that fulfilling the three psychological needs can promote intrinsic motivation. This argument makes sense when we talk about conditional change in SR. That is, a child can temporarily perform better in a context where the three needs are fulfilled, but changing the condition would also change the SR performance. Such a change in SR is temporary and reversible. However, the goal of intervention is long-term changes in SR regardless of condition and context. I recommend the author elaborate on how conditional change in SR can lead to long-term change in SR as a developmental outcome.
The author pointed out that previous reviews failed to include SR intervention in out-of-school contexts. However, the synthesis of the results only included analysis of the autonomy, relatedness, and competence components. To echo the study rationale, analysis of the intervention contexts should also be included. The role of the three psychological needs can potentially be moderated by the intervention contexts.
The context was intended as an initial analysis item but only one study was conducted in the home and one in the lab. All remaining studies were done in the formal learning context. As such, there was not an opportunity to analyse by context. We have added a sentence to this effect, as follows:
Analysis of context as a potentially differentiating characteristic was not possible given there was one study in the home learning context, one in the lab, and all the remaining studies in the formal learning context. (p. 6).
I am also worried that the search terms used for the search were not inclusive enough and many relevant studies were not included. For example, self-regulation during preschool years is often conceptualized and measured as executive functions and school readiness, but these two terms were not included. Subcomponents of executive functions (e.g., inhibitory control and cognitive flexibility) are sometimes used as the outcome in the studies without explicitly using the word “regulation”. For example, Head Start REDI and the Chicago School Readiness Project are two intervention programs that target preschoolers’ self-regulation, some of the publications even tested longitudinal outcomes in follow-up studies. Moreover, Head Start REDI included home components in later development. These two intervention programs use several curriculums, including Tools of Mind, The Incredible Years, and the PATH. Here are some publications between 2010 and 2020 that are relevant but not included:
Raver, C. C., Jones, S. M., Li-Grining, C., Zhai, F., Bub, K., & Pressler, E. (2011). CSRP's impact on low-income preschoolers' preacademic skills: Self-regulation as a mediating mechanism. Child Development, 82(1), 362-378.
Jones, S. M., Bub, K. L., & Raver, C. C. (2013). Unpacking the black box of the Chicago School Readiness Project intervention: The mediating roles of teacher–child relationship quality and self-regulation. Early Education & Development, 24(7), 1043-1064. doi:10.1080/10409289.2013.825188
Li-Grining, C., Raver, C., Jones-Lewis, D., Madison-Boyd, S., & Lennon, J. (2014). Targeting classrooms' emotional climate and preschoolers' socioemotional adjustment: Implementation of the Chicago School Readiness Project. Journal of Prevention and Intervention in the Community, 42(4). 264-281
Zhai, F., Raver, C., Jones, S. M., Li-Grining, C. P., Pressler, E., & Gao, Q. (2010). Dosage effects on school readiness: Evidence from a randomized classroom-based intervention. Social Service Review, 84(4), 615-654.
Zhai, F., Raver, C. C., & Li-Grining, C. (2011). Classroom-based interventions and teachers’ perceived job stressors and confidence: Evidence from a randomized trial in Head Start settings. Early Childhood Research Quarterly, 26, 442-452. doi:10.1016/j.ecresq.2011.03.003
Zhai, F., Raver, C. C., & Jones, S. M. (2012). Academic performance of subsequent schools and impacts of early interventions: Evidence from a randomized controlled trial in Head Start settings. Children and Youth Services Review, 34, 946-954. doi:10.1016/j.childyouth.2012.01.026
Bierman, K. L., Nix, R. L., Heinrichs, B. S., Domitrovich, C. E., Gest, S. D., Welsh, J. A., & Gill, S. (2014). Effects of Head Start REDI on children's outcomes 1 year later in different kindergarten contexts. Child Development, 85(1), 140-159. doi:10.1111/cdev.12117
Nix, R. L., Bierman, K. L., Heinrichs, B. S., Gest, S. D., Welsh, J. A., & Domitrovich, C. E. (2016). The randomized controlled trial of Head Start REDI: Sustained effects on developmental trajectories of social-emotional functioning. Journal of Consulting and Clinical Psychology, 84(4). 310-322.
Sasser, T. R., Bierman, K. L., Heinrichs, B., & Nix, R. L. (2017). Preschool intervention can promote sustained growth in the executive-function skills of children exhibiting early deficits. Psychological Science, 28(12), 1719-1730.
Bierman, K. L., Welsh, J. A., Heinrichs, B. S., Nix, R. L., & Mathis, E. T. (2015). Helping Head Start Parents promote their children's kindergarten adjustment: The research-based developmentally informed parent program. Child Development, 86(6), 1877-1891. doi:10.1111/cdev.12448
Bierman, K. L., Heinrichs, B. S., Welsh, J. A., Nix, R. L., & Gest, S. D. (2017). Enriching preschool classrooms and home visits with evidence-based programming: Sustained benefits for low-income children. Journal of Child Psychology And Psychiatry, 58(2), 129-137.
Moore, J. E., Cooper, B. R., Domitrovich, C. E., Morgan, N. R., Cleveland, M. J., Shah, H., & ... Greenberg, M. T. (2015). The effects of exposure to an enhanced preschool program on the social-emotional functioning of at-risk children. Early Childhood Research Quarterly, 32, 127-138. doi:10.1016/j.ecresq.2015.03.004
Shah, H. K., Domitrovich, C. E., Morgan, N. R., Moore, J. E., Cooper, B. R., Jacobson, L., & Greenberg, M. T. (2017). One or two years of participation: Is dosage of an enhanced publicly funded preschool program associated with the academic and executive function skills of low-income children in early elementary school? Early Childhood Research Quarterly, 40, 123-137.
With regard to the conceptual distinction between self-regulation (SR) and executive function (EF), although there is conceptual overlap and some (perhaps unadvised) interchangeable use of these terms, there is evidence that these are distinct abilities. EF is not SR, per se, although it likely contributes to it. This can also be said for school readiness, that while SR contributes to this ability, it is not in and of itself SR. While it is conceivable that there are EF or school readiness interventions that instead (or also) target aspects aligned to our conception of SR, our focus here was on prior-to-school interventions that privilege SR as its intervention target. It is for this reason that we did not expand our search terms to capture related yet conceptually distinct aspects such as executive function, school readiness, temperament, social-emotional wellbeing, etc.
Thank you as well for providing this list of articles. Each actually turned up in our search results but were excluded for one or more reasons in relation to our inclusion/exclusion criteria. However, this comment helped us realise that perhaps these were not as explicitly articulated as we had thought. Specifically, as noted earlier in the manuscript and applied to our study selection process (but not further articulated in our inclusion criteria section), there was a need for studies to provide sufficient detail to map against SDT components. This required at least some cursory specification of children’s activity as a consequence of intervention implementation. This has now been added also to the inclusion criteria section. (p. 5), and also to the title.
There were a number of studies that were excluded on the basis of this now-better-articulated requirement. This was commonly for studies that involved educator professional development yet did not specify child activity that educators should stimulate or implement their new learnings within. As such, it was impossible to classify the following studies in terms of whether (or not) they targeted child autonomy, competence, and/or relatedness.
- Raver, C. C., Jones, S. M., Li-Grining, C., Zhai, F., Bub, K., & Pressler, E. (2011). CSRP's impact on low-income preschoolers' preacademic skills: Self-regulation as a mediating mechanism. Child Development, 82(1), 362-378. & and all CSRP papers
- Jones, S. M., Bub, K. L., & Raver, C. C. (2013). Unpacking the black box of the Chicago School Readiness Project intervention: The mediating roles of teacher–child relationship quality and self-regulation. Early Education & Development, 24(7), 1043-1064. doi:10.1080/10409289.2013.825188
- Li-Grining, C., Raver, C., Jones-Lewis, D., Madison-Boyd, S., & Lennon, J. (2014). Targeting classrooms' emotional climate and preschoolers' socioemotional adjustment: Implementation of the Chicago School Readiness Project. Journal of Prevention and Intervention in the Community, 42(4). 264-281.
- Zhai, F., Raver, C. C., & Jones, S. M. (2012). Academic performance of subsequent schools and impacts of early interventions: Evidence from a randomized controlled trial in Head Start settings. Children and Youth Services Review, 34, 946-954. doi:10.1016/j.childyouth.2012.01.026
We have also added text to the limitations section, citing a number of these studies, to indicate that this leaves open the possibility that other approaches (that cannot be mapped against SDT) might be equally or more effective than those reviewed here, and/or that characteristics of effective intervention may differ for these disparate approaches. We have added the following text to address this:
Interventions that could not be classified according to SDT (eg Raver et al., 2011; Li-Grinning et al., 2014) could not be considered given the aim of this review. However, this leaves open the possibility that other approaches might be equally (or even more) effective, or that components that increase effectiveness might differ for other approaches (e.g., professional development approaches that do not specify child activity, but rather intervene within it). (p. 13-14.)
Another set of studies were excluded because they did not specifically measure SR as an outcome of the intervention, which was a stated inclusion criterion. While many of these studies included items or subscales that could be construed as SR (while others included EF measures), they were then aggregated with items/subscales of related but discrete abilities (e.g., prosocial behaviour) – thus impact of the intervention on SR, per se, could not be evaluated. This was the case for the following studies:
- Zhai, F., Raver, C., Jones, S. M., Li-Grining, C. P., Pressler, E., & Gao, Q. (2010). Dosage effects on school readiness: Evidence from a randomized classroom-based intervention. Social Service Review, 84(4), 615-654.
- Zhai, F., Raver, C. C., & Li-Grining, C. (2011). Classroom-based interventions and teachers’ perceived job stressors and confidence: Evidence from a randomized trial in Head Start settings. Early Childhood Research Quarterly, 26, 442-452. doi:10.1016/j.ecresq.2011.03.003
- Bierman, K. L., Nix, R. L., Heinrichs, B. S., Domitrovich, C. E., Gest, S. D., Welsh, J. A., & Gill, S. (2014). Effects of Head Start REDI on children's outcomes 1 year later in different kindergarten contexts. Child Development, 85(1), 140-159. doi:10.1111/cdev.12117
- Nix, R. L., Bierman, K. L., Heinrichs, B. S., Gest, S. D., Welsh, J. A., & Domitrovich, C. E. (2016). The randomized controlled trial of Head Start REDI: Sustained effects on developmental trajectories of social-emotional functioning. Journal of Consulting and Clinical Psychology, 84(4). 310-322.
- Bierman, K. L., Welsh, J. A., Heinrichs, B. S., Nix, R. L., & Mathis, E. T. (2015). Helping Head Start Parents promote their children's kindergarten adjustment: The research-based developmentally informed parent program. Child Development, 86(6), 1877-1891. doi:10.1111/cdev.12448
- Bierman, K. L., Heinrichs, B. S., Welsh, J. A., Nix, R. L., & Gest, S. D. (2017). Enriching preschool classrooms and home visits with evidence-based programming: Sustained benefits for low-income children. Journal of Child Psychology And Psychiatry, 58(2), 129-137.
- Moore, J. E., Cooper, B. R., Domitrovich, C. E., Morgan, N. R., Cleveland, M. J., Shah, H., & ... Greenberg, M. T. (2015). The effects of exposure to an enhanced preschool program on the social-emotional functioning of at-risk children. Early Childhood Research Quarterly, 32, 127-138. doi:10.1016/j.ecresq.2015.03.004
- Sasser, T. R., Bierman, K. L., Heinrichs, B., & Nix, R. L. (2017). Preschool intervention can promote sustained growth in the executive-function skills of children exhibiting early deficits. Psychological Science, 28(12), 1719-1730.
- Shah, H. K., Domitrovich, C. E., Morgan, N. R., Moore, J. E., Cooper, B. R., Jacobson, L., & Greenberg, M. T. (2017). One or two years of participation: Is dosage of an enhanced publicly funded preschool program associated with the academic and executive function skills of low-income children in early elementary school? Early Childhood Research Quarterly, 40, 123-137
Moreover, some studies included more than one component, and the effect of the inclusion of components may not be additive. The combination of certain components may yield a desirable result while each component cannot produce an effect on its own.
Thank you for this observation. The complex interaction of components has been recognised in the conclusion, but this comment highlights the need for further discussion. The conclusion has been modified to highlight this point:
While the review has thus far assumed that autonomy may play a supportive role alongside competence or relatedness, it may be the case that effects are not additive. Various combinations may increase or decrease efficacy in a non-additive manner, for instance, with a single SDT factor generating additional benefit when combined with one or more other factors. The fact that no single study included autonomy in isolation means we cannot draw conclusions as to the effect of this component on its own. This possibility requires further research to investigate. (p. 14).
Reviewer 2 Report
The article deals with an important issue concerning the effectiveness of interventions aimed at the development of "Self-Regulation" (SR) in preschool age. The authors seem to recognize SR as the child's ability to modulate behavior towards achieving a goal in the child's developmental and educational context. The authors reviewed the literature on this subject, guided by clearly defined criteria, which were related to the Self-Determination Theory adopted as the basis for understanding the essence and meaning of self-regulation. The purpose of analyzing the reports contained in the articles published in 2010-2020 was to identify the most effective intervention approaches (programs, activities, people conducting the intervention, etc.).
Their findings indicate that targeting the interventions on shaping competencies through encouragement and feedback, and stimulating children's autonomy is more effective from the point of view of self-regulation in preschool-age development than the involvement of relatives (child caregivers, educators, teachers) in intervention activities. Play activity seems to be particularly effective for the development of self-regulation. Focusing on play is justified from the perspective of developmental psychology in the fact that it is a basic activity during childhood.
The authors indicate the limitations of their analyzes and the necessity/usefulness of further research-focused, inter alia, on the family environment.
Author Response
Thank you so much for your great summary and positive evaluation.
Reviewer 3 Report
First, thank you for allowing me to read this paper which initially deals with an interesting and relevant topic. The suggestions given in this document are intended to improve your work. The same feedback document will be given to both editors and authors.
Title:
- Remove systematic from the title, as it can be confusing.
Introduction section:
- The first part of the introduction also needs a title as it is a sub-section.
- Line 72. Such phrases are not necessary. Remember that this is a scientific text: "These will be described in the next section". Please check the text.
- I miss some explanation about the development of SR.
- Lines 126-128. This is information that should be in the methods section.
Methods:
- Some information on the method is missing. I recommend follow the methodological recommendations of Arksey and O'Malley and Levac et al. I suggest authors review them:
- Arksey, H.; O’Malley, L. Scoping studies: Towards a methodological framework. Int J Soc Res Method 2005, 8, 19–32.
- Levac, D.; Colquhoun, H.; O’Brien, K.K. Scoping studies: Advancing the methodology. Implement Sci 2010, 5, 69. 467
- Scoping reviews do not have to include quality and risk of bias assessment. But since you influence the first, the ideal is to use one of the many tools available.
- I think a review of the PRIZMA methodology checklist can help make the work even better. http://www.prisma-statement.org/Extensions/ScopingReviews
Results:
- The table attached as supplementary material is important enough to be included in the original text. Remember not to duplicate in the text information that is in tables or figures.
- Include the flow chart considering the PRISMA statement recommendations. http://www.prisma-statement.org/Extensions/ScopingReviews
Discussion:
- I think it is relevant and well structured. However, it may include the limitations of scoping reviews (i.e. what differentiates it from systematic reviews).
- Do you have any future directions for this work? Practical implications should be placed in this section.
Conclusions:
- The conclusion should refer to your results, answering your research question concisely. Please rewrite this section.
Author Response
Responses to comments of Reviewer 3 (Reviewer’s comments are printed in boldface)
Title:
Remove systematic from the title, as it can be confusing.
There are growing calls to also characterise scoping reviews as systematic, especially given they follow the systematic methods of other review types. This is per the published guidance from the Joanna Briggs Institute – a leading organisation in the field of reviews – below:
https://pubmed.ncbi.nlm.nih.gov/26134548/
The methodology of our review fulfils these stated criteria to be deemed a systematic scoping review.
Introduction section:
The first part of the introduction also needs a title as it is a sub-section.
As suggested, the sub-heading “1.1 Background” has been added (p. 1)
Line 72. Such phrases are not necessary. Remember that this is a scientific text: "These will be described in the next section". Please check the text.
This sentence has been removed. (p. 2) The full text was also screened for any additional similar instances and were removed when found. (p. 8 and p. 11 on original manuscript).
I miss some explanation about the development of SR.
The following text has been added to briefly characterise typical SR development in early childhood:
It is largely accepted that children experience a period of rapid SR development between the ages of 3 and 7 years, and the nonlinear way in which this occurs results in different growth trajectories (Diamond, 2002). Contributing to these disparities, both biological predisposition and environmental experience – which vary by child, family and circumstance – influence early SR development (Montroy et al., 2016). (p. 1).
Lines 126-128. This is information that should be in the methods section.
These lines have been moved to section 2.7 as suggested. (p. 6)
Methods:
Some information on the method is missing. I recommend follow the methodological recommendations of Arksey and O'Malley and Levac et al. I suggest authors review them:
- Arksey, H.; O’Malley, L. Scoping studies: Towards a methodological framework. Int J Soc Res Method 2005, 8, 19–32.
- Levac, D.; Colquhoun, H.; O’Brien, K.K. Scoping studies: Advancing the methodology. Implement Sci 2010, 5, 69. 467
Thank you for providing the citations mentioned. We did indeed read and consult these during conception of the project, but considered the more recent PRISMA-ScR (Tricco et al., 2018) guidelines as a more suitable framework to guide our review, given that it draws on and further expands on these previous works. The PRISMA-ScR was consulted throughout the process and write-up of this paper.
Scoping reviews do not have to include quality and risk of bias assessment. But since you influence the first, the ideal is to use one of the many tools available.
The PRISMA-ScR refers to this as a Critical Appraisal of individual sources of evidence (item 12 on the published checklist) “instead of a risk of bias” (page 2 in key section of checklist), and thus a tool to assess risk of bias is not necessary, as you have mentioned. As such, we have opted not to include this as it is not a typical feature of scoping reviews – which seek to describe and reconcile a new or highly disparate field of research.
I think a review of the PRIZMA methodology checklist can help make the work even better. http://www.prisma-statement.org/Extensions/ScopingReviews
The checklist of the PRISMA-ScR items has been completed and has been included as an Appendix (pending journal confirmation that this can be supported).
Results:
The table attached as supplementary material is important enough to be included in the original text. Remember not to duplicate in the text information that is in tables or figures.
The table is too large to be formatted as per the journal requirements, hence why it was initially submitted as supplementary material. That said, if the journal can support moving the table into the body of the article, we are happy for its inclusion there. We will consult with the journal to see what is possible. We suggest placement on p. 8 after Figure 1.
Include the flow chart considering the PRISMA statement recommendations. http://www.prisma-statement.org/Extensions/ScopingReviews
The flow chart (item 14 on the PRISMA-ScR) has been included in the manuscript. (p. 7)
Discussion:
I think it is relevant and well structured. However, it may include the limitations of scoping reviews (i.e. what differentiates it from systematic reviews).
Do you have any future directions for this work? Practical implications should be placed in this section.
Conclusions:
The conclusion should refer to your results, answering your research question concisely. Please rewrite this section.
…this scoping review sought to identify characteristics of (more, less or in-) effective pre-school SR interventions from a Self-Determination Theory (SDT) perspective. (p. 2)
And
In relation to the research question guiding this review, the findings suggest that content factors (what interventions did) and implementation design factors (how interventions executed them) interacted to account for differences in the presence and size of intervention effects. Using the SDT framework guiding this review, the results pointed to competence being a core driver for SR growth particularly when encouragement and feedback was provided. What remains unclear, however, is whether this experience of competence supported children to perform to their full SR abilities and/or whether it instigated growth in SR. Nevertheless, given optimal self-regulatory function is an important condition for self-regulation growth (REF), these are beneficial improvements compared to business-as-usual controls. Autonomy and relatedness seemed to complement efforts to target competence, but future research needs to identify ways in which they can be meaningfully integrated to support the growth of each of these components. (p. 14)